# Technical Performance of a 430-Gene Preventative Genomics Assay to Identify Multiple Variant Types Associated with Adult-Onset Monogenic Conditions, Susceptibility Loci, and Pharmacogenetic Insights

**DOI:** 10.3390/jpm12050667

**Published:** 2022-04-21

**Authors:** Ari Silver, Gabriel A. Lazarin, Maxwell Silver, Meghan Miller, Michael Jansen, Christine Wechsberg, Erin Dekanek, Stav Grossfeld, Tim Herpel, Dinura Gunatilake, Alexander Bisignano, Malgorzata Jaremko

**Affiliations:** Phosphorus, Inc., 1140 Broadway, 12th Floor, New York, NY 10001, USA; gabriel@phosphorus.com (G.A.L.); maxwell@phosphorus.com (M.S.); meghan@phosphorus.com (M.M.); mike@phosphorus.com (M.J.); christine@phosphorus.com (C.W.); erin@phosphorus.com (E.D.); stav.grossfeld@phosphorus.com (S.G.); tim@phosphorus.com (T.H.); dinuragunatilake@phosphorus.com (D.G.); alex@phosphorus.com (A.B.); gosia@phosphorus.com (M.J.)

**Keywords:** genetics, population health, bioinformatics, next generation sequencing (NGS), genomic screening

## Abstract

DNA-based screening in individuals without known risk factors potentially identifies those who may benefit from genetic counseling, early medical interventions, and/or avoidance of late or missed diagnoses. While not currently in widespread usage, technological advances in genetic analysis overcome barriers to access by enabling less labor-intensive and more cost-efficient means to discover variants of clinical importance. This study describes the technical validation of a 430-gene next-generation sequencing based assay, GeneCompass^TM^, indicated for the screening of healthy individuals in the areas of actionable health risks, pharmaceutical drug response, and wellness traits. The test includes genes associated with Mendelian disorders and genetic susceptibility loci, encompassing 14 clinical areas and pharmacogenetic variants. The custom-designed target enrichment capture and bioinformatics pipelines interrogate multiple variant types, including single nucleotide variants, insertions/deletions (indels), copy number variants, and functional haplotypes (star alleles), including tandem alleles and structural variants. Validation was performed against reference DNA from three sources: 1000 Genomes Project (*n* = 3), Coriell biobank (*n* = 105), and previously molecularly characterized biological specimens: blood (*n* = 15) and saliva (*n* = 11). Analytical sensitivity and specificity for single nucleotide variants (SNVs) were 97.57% and 99.99%, respectively, and for indels were 74.57% and 97.34%, respectively. This study demonstrates the validity of an NGS assay for genetic screening and the broadening of access to preventative genomics.

## 1. Introduction

DNA-based screening for adult health conditions aims to guide preventative care for individuals without apparent medical indications for the condition(s) being screened [1]. This is differentiated from other large-scale DNA screening protocols, such as carrier screening for prospective parents, or aneuploidy screening during pregnancy, but with similar aims. Public-health-based screening criteria was first set forth by Wilson and Jungner in 1968, which informed the Centers for Disease Control and Prevention (CDC) establishment of “tier 1” conditions suitable for population-based screening, which include 10 genes associated with breast and ovarian cancer syndrome, Lynch syndrome, and familial hypercholesterolemia [2,3]. Second tier gene–condition pairs consist of pharmacogenetic traits and certain monogenic conditions, while the third tier encompasses polygenic risk scores and gene–environment pairings [3]. While these recommendations were made by the CDC regarding genes and conditions to be screened for, testing methodology was and has not since been specified.

Few initiatives currently exist to deliver genomic screening beyond these 10 genes at a population scale. For example, the MyCode Community Health Initiative (Geisinger Health. https://www.geisinger.org/precision-health/mycode. accessed on 16 February 2022) and the Genomic Health Initiative (NorthShore University Health System. https://www.northshore.org/personalized-medicine/research-innovation/genomic-health-initiative/. accessed on 16 February 2022) represent localized efforts. These programs are funded as research studies and participants receive data if actionable findings are identified, as deemed by the researchers. However, to date, the majority of individuals in the United States have limited access to genetic health screening outside the context of a research study due to multiple barriers. Some of these barriers may include the current absence of endorsement by important and influential organizations, like the American College of Medical Genetics and Genomics for population-based screening initiatives.

Despite limited access to genetic screening, evolving data on population-based genetic screening suggests clinical utility and benefits. The Healthy Nevada Project reported that 90% of carriers for the CDC tier one conditions had not previously been identified by the health system and 75% did not report a related family history [4]. The MyCode initiative reported a 0.8% prevalence of pathogenic copy number variants (CNVs) that are causative of a neuropsychiatric disorder. Despite 70% of these individuals being symptomatic, less than 6% had previously received a genetic diagnosis. Treatment was symptom-based without a diagnostic understanding of the underlying etiology [5].

This data also demonstrates the public interest in receiving genomic test results, including information well beyond those that meet the highest evidence thresholds. Perceived disease severity was sufficient to motivate interest in receiving genetic information [6] and whole genome sequencing results did not cause long-term anxiety or depression, but did motivate behavioral changes, and could be managed appropriately in primary care settings [7]. These results indicate a potential readiness for deployment of genetic screening at a greater scale than currently exists.

While understanding of the optimal implementation and outcomes of genetic health screening will continue to evolve, current data support that the expansion of access to a broader array of genetic information may benefit individuals and their overall health care. This study describes data that validate GeneCompass, a 430-gene next-generation sequencing-based (NGS) assay that identifies variants associated with monogenic diseases, susceptibility to complex conditions, drug response, and wellness traits, with the purpose of providing a medically actionable, affordable, and scalable personalized genetic screening tool. The next-generation sequencing-based platform discovers multiple categories of variants, including copy number variants and star alleles, with the aim of balancing actionability, severity, comprehensiveness, accuracy, and cost. While this assay expands to a broader set of clinical areas, it still maintains high accuracy for SNVs and indels of 99.92% and 94.77%, respectively.

## 2. Materials and Methods

### 2.1. Samples

The DNA was extracted from saliva and whole blood specimens using the Maxwell^®^ RSC Blood DNA Kit (Promega, Madison, Wisconsin) on the automated Maxwell^®^ RSC 48 Instrument (Promega), as well as being sourced from the Coriell Cell Repository. The validation set included 157 samples: (1) DNA from: 1000 Genomes Project [8] (*n* = 3), Coriell (*n* = 105), and CAP proficiency samples (*n* = 2); and (2) previously molecularly characterized clinical saliva (*n* = 11) and whole blood samples (*n* = 15). The samples were selected based on the absence/presence of known pathogenic/likely pathogenic variants in the genes included on the panel, prior pharmacogenetic characterization for specific star allele and non-star allele genes, and the availability of the material for the clinical samples. The clinical samples were fully blinded for the purposes of this validation and were handled according to the Health Insurance Portability and Accountability Act standards, and the internal samples were derived from individuals who had consented to the use of their sample for de-identified research as a secondary purpose at the time of their original clinical test request. Appendix A describes all of the samples used in this study, with biorepository numbers, as well as previous variant characterizations and the rationale for inclusion.

### 2.2. Gene Selection and Rationale

The GeneCompass assay comprises four segments: two pertaining to health risks with Mendelian inheritance (monogenic disease) and non-Mendelian inheritance (susceptibility loci), pharmacogene interactions, and wellness traits. The assay content is summarized in Table 1 below and is detailed by gene, condition, and targeted variants, where applicable, in the Appendix A. Rationale for gene/targeted variant inclusion were: primary research evidence that supports a gene–disease association, as well as evidence documented through the guidelines and public databases (ClinGen and ClinVar), genes currently available for testing through other clinical laboratories, recent emergent or suggestive evidence of an association with certain conditions (e.g., COVID-19 susceptibility or severity), pharmacogenetic associations described by the Clinical Pharmacogenetics Implementation Consortium and/or the Pharmacogenomics Knowledge Base, and/or interest by health care providers, as well as wellness traits described in GWAS studies. Wellness variants were included based on the results of genome-wide association studies and their clinical relevance.

The first test segment includes conditions primarily of monogenic etiology and with adult-onset phenotypes. Pathogenic variants in the genes associated with these conditions are highly predictive of phenotypes, although there may be a variable spectrum of manifestations. There are 430 unique genes encompassing approximately 14 clinical areas, defined broadly by either the affected organ system or the type of disease, as seen in Table 1. In terms of gene content, the three largest areas are cardiovascular disease (95 genes, 50 conditions), oncology (70 genes, 57 conditions), and reproductive health (43 genes, 26 conditions). Some genes may be associated with more than one condition or clinical area and some gene data is captured, but not reported. In general, this test segment is assessed using next-generation sequencing with full exon, and +/− 12 intron–exon junction interpretations. For certain genes, copy number variant analysis is added to maximize the clinical sensitivity. The results are reported as either positive or negative, based on the identification of a pathogenic or likely pathogenic variant. As a screening test, variants of uncertain significance are not reported as a default.

Mendelian and non-Mendelian disease penetrance differentiate the first and second segments. The latter, “susceptibility loci”, located in 67 genes, positively or negatively modify the risk for a given condition, and are reported in the same disease category sections as monogenic disorders. As an example of the difference, the cardiovascular disease area includes familial hypertrophic cardiomyopathy, an autosomal dominant condition caused by pathogenic variants in the *MYH7*, *MYBPC3*, *TNNT2*, and *TNNI3* genes. It also includes a risk assessment for myocardial infarction by evaluating the *IL4* gene for the c.-589C>T (rs2243250) susceptibility variant. This variant in its heterozygous and homozygous state has been described as reducing the occurrence of myocardial infarction [9].

As the research and the literature documenting susceptibility loci are often specific to one or few variants within a given gene, the GeneCompass test specifically targets these variants of interest and does not currently assess other variants within those genes (except when relevant for monogenic conditions). After the determination of pathogenicity of the variant, the results are further stratified by classifying the variant via a risk rating system. This system provides clarification on the phenotypic risk level associated with the susceptibility loci.

The pharmacogenetic portion targets specified variants in 122 genes known to modify the response to pharmaceutical medications across all of the previously described clinical areas. As some genes are informative for more than one drug or drug class, 349 total drug response possibilities are investigated. Variants were chosen based on guidance issued by the United States Food and Drug Administration or evidence levels from the PharmGKB and/or Clinical Pharmacogenetic Implementation Consortium databases, as well as the currently available literature. The results are reported according to three categorizations: use as directed; use with caution; or use with extreme caution/consider alternatives. Use as directed indicates that the patient has the typical genotype and the provider can proceed with prescribing as needed. Use with caution indicates that the patient’s genotype shows that they may not have an ideal response to treatment, as demonstrated by reduced or altered efficacy or side effects. A result showing “use with extreme caution/consider alternatives” demonstrates that the patient’s genotype is associated with adverse outcomes that may impact patient safety.

The wellness test segment assesses 46 genetic traits that, while not intended to guide the diagnosis of medical conditions, may affect holistic wellness by, for example, modifying the body’s metabolic or sleep processes. Results are listed across four categories: sleep; diet and nutrition; fitness and exercise; and skin and haircare. Within each category, results are reported describing the potential trait–genotype effect and are stated to not hold equivalence with disease-associated results.

The amount of resultant information necessitates clear prioritization and explanation to the clinician and individual. Consequently, findings are ranked based on the assigned risk rating, to emphasize the highest risks for the most severe conditions. These rankings dictate the need for medical management and genetic counseling protocols; further research will validate the agreement and usability of them. All patients, in particular those that receive a rating of “high risk,” are encouraged to pursue genetic counseling. Genetic counseling services are a component in ensuring that patients have the tools to understand the impact of their results and the actionable insights from this test. Genetic counselors provide strategic plans for patients to move forward with an understanding of their risk and how to mitigate disease manifestation.

### 2.3. Standard Variant Workflow

#### 2.3.1. Next-Generation Sequencing

The NGS library preparation utilized a custom-designed targeted capture of selected genes/variants and followed the Fast Hybridization Target Enrichment workflow (Twist Bioscience, South San Francisco, CA). The samples were prepared using 50 ng of input DNA and the Twist EF Library Prep Kit 2.0 + UDIs and Fast Hyb reagents (Twist Bioscience) and run in a format of 10 samples per pool and 30 per single run on a MiniSeq System (Illumina, San Diego, CA, USA).

The capture is designed based on the human reference genome assembly GRCh37 (Genome Reference Consortium Human Build 37) and fully covers sequenced gene exonic regions with +/− 12 nucleotides in the exon–intron junctions and any intronic pathogenic variants as defined by ClinVar, as well as targeted intervals related to targeted susceptibility, pharmacogenetics, and wellness loci. Clinically significant variants reportable for monogenic conditions were confirmed using Sanger sequencing for single nucleotide variants (SNVs), indels, and small deletions and insertions. Genes containing CNVs were confirmed as described below.

A set of standard NGS quality metrics are calculated for each sample processed, including total reads, average depth, uniformity of coverage (the percent of targeted bases with a read depth greater than 0.2 times the mean depth of the sample), on-target percentage, duplicate read percentage, and percent of a base read depth greater than 20×. These metrics are determined by the Edico software version 3.4.5 (Illumina). If a sample is below our minimum required threshold for any metric, the sample is re-sequenced to obtain sufficient data quality.

#### 2.3.2. Bioinformatics Pipeline

The NGS runs produce binary base call (BCL) files for all samples that are processed, which are automatically uploaded to Basespace (Illumina Corporation, San Diego, CA, USA) and converted into pair-end FASTQ files. Next, the Edico software version 3.4.5 (Illumina) performs an alignment to convert these pair-end FASTQ files into binary alignment map (BAM) files for each sample. Concurrently, the Edico software also performs variant calling to produce a variant call format (VCF) file for each sample, which it then combines using joint-processing to create a single joint VCF file for the batch. Based on the Genome Analysis Tool Kit’s (GATK) algorithms, the Dragen Edico server performs rapid alignments using a Field Programmable Gate Arrays (FPGA) chip, which allows the software to perform alignments and variant calling for seven–eight samples per hour.

In addition to each sample’s VCF file, an artificially created VCF containing relevant reference positions, in which the allele of interest is the reference allele defined by the GRCh37 reference sequence, is also created to be included as part of the joint processing so that the sequencing information for these positions can be included, even if none of the samples in the batch have a non-reference call at the position. This allows for treating the reference allele in a parallel fashion to any other allele. It also enables distinguishing between a missed call in the VCF due to a subject being wild-type, compared to a missed call related to a lack of coverage.

The joint VCF file is then processed to upload individual variant results for each sample into the database. For each variant call, the depth and quality of the call, as well as strand depth information, is processed and stored. This information is used to approve or disapprove each call. Only approved calls are used for downstream processing.

#### 2.3.3. Variant Annotation and Classification

For all new mutations that are created from any pipeline in the process, a VCF file is generated, in which there is no subject information stored, but instead just the genomic coordinates and alleles. This VCF file is then annotated with both RefSeq’s SnpEff V4.3 and Ensembl’s Variant Effect Predictor (VEP v94) to produce annotations for each provided mutation, which are stored in the database [10,11]. Pathogenicity assessment for variants in genes tested for monogenic disorders was assessed based on the standards and guidelines published by the American College of Medical Genetics and Genomics (ACMG) [1]. The assessment followed the ACMG evidence framework (including the population, computational, functional, segregation, allelic, and other data) and classification criteria to interpret variants using a five-tier classification system consisting of variants called: pathogenic; likely pathogenic; uncertain significance; likely benign; and benign. Additionally, the gene-specific interpretation guidelines were published by ClinGen panel experts.

#### 2.3.4. Variant Validation Analysis

The performance characteristic of the assay was established by the assessment of accuracy, analytical sensitivity and specificity, precision, detection rates of targeted alleles, reportable range, and quality of the NGS metrics produced by two types of biological samples included in analysis. The assay performance was defined by the following five key metrics:(1)$$\mathrm{Accuracy}=\;\frac{\mathrm{TP}+\mathrm{TN}}{\left(\mathrm{TP}+\mathrm{FP}+\mathrm{TN}+\mathrm{FN}\right)}$$
(2)$$\mathrm{Sensitivity}=\;\frac{\mathrm{TP}}{\mathrm{TP}+\mathrm{FN}}$$
(3)$$\mathrm{Specificity}=\;\frac{\mathrm{TN}}{\mathrm{FP}+\mathrm{TN}}$$
(4)$$\mathrm{Positive}\;\mathrm{Predictive}\;\mathrm{Value}=\;\frac{\mathrm{TP}}{\mathrm{TP}+\mathrm{FP}}$$
(5)$$\mathrm{Negative}\;\mathrm{Predictive}\;\mathrm{Value}=\;\frac{\mathrm{TN}}{\mathrm{TN}+\mathrm{FN}}$$

Accuracy was measured as a concordance of any non-reference call, including variants of all pathogenicity, to the pilot mask of the 1000 Genome Project. Intra- and inter-precision studies utilized three 1000 Genomes Project samples run in triplicate in a single run and across three different runs, respectively. These runs were performed by different operators. Precision was assessed based on the reproducibility of the following NGS metrics: aligned reads (%); duplicates (%); on target (%); average base depth per sample on reportable gene regions; bases with depth ≥ 20× (%); average depth per reportable interval; and % of intervals with depth ≥ 20×. Reproducibility was assessed on the replicates of the same samples by counting concordant genotype calls if all three replicates of non-reference genotypes in a sample agreed.

The detection rate of our targeted alleles was compared to the population frequency listed in the Genome Aggregation Database (gnomAD) to determine if our sample variant frequencies are in concordance with the variant frequencies in the population [12]. The gnomAD database has variant data on 125,748 exomes and 15,708 genomes from unrelated individuals sequenced against the GRCh37 reference sequence. Frequencies on single nucleotide polymorphisms (SNP) and indels were pulled from gnomAD on 10 January 2022. Concordance was measured by performing a two-sample equal variance *t*-test on each section of the targeted alleles in our dataset.
(6)$$S^{2}=\;\frac{\sum_{i=1}^{n_{1}}{\left(x_{i}-{\bar{x}}_{1}\right)}^{2}+\sum_{j=1}^{n_{2}}{\left(x_{j}-{\bar{x}}_{2}\right)}^{2}}{n_{1}+n_{2}-2}$$
(7)$$t_{statistic}=\;\frac{{\bar{x}}_{1}-{\bar{x}}_{2}}{\sqrt{S^{2}\left(\frac{1}{n_{1}}+\frac{1}{n_{2}}\right)}}$$

The statistical output was then evaluated against the null hypothesis of our sample population being equal to the general population.

### 2.4. Specialized Variant Workflow

#### 2.4.1. Copy Number Variants

We detect genes containing large CNVs from targeted NGS sequencing data using GATK’s CNV Caller [13]. This is performed at the gene-level for targeted exome data to reach a high confidence for the reportable variants. We provide a list of genomic intervals of length 300 bp to initiate *CollectReadCounts*, which is the first step in the pipeline. Determining an appropriate bin size is critically important as a larger bin size may discount smaller CNVs as noise, while a smaller bin size may overcount CNVs due to normal coverage variations in the data. This tool tabulates the raw integer counts of reads overlapping an interval on each sample BAM file. The outputted counts move into *DetermineGermlineContigPloidy* to call contig level ploidies for both autosomal and allosomal contigs. Baseline contig ploidies are determined through sample coverages and contig ploidy priors that give the prior probabilities for each ploidy state for each contig. In this step, the tool generates global baseline coverage and noise data that will be utilized during the model creation downstream.

The *GermlineCNVCaller* learns a denoising model per scattered shard while consistently calling CNVs across the shards. The tool models systematic biases and CNVs simultaneously, which allows for sensitive detection of both rare and common CNVs. *PostprocessGermlineCNVCall* is used to consolidate the scattered *GermlineCNVCaller* results, perform segmentation, and call the copy number states. This tool generates per-interval and per-segment sample calls in VCF format for each sample processed. The final step deployed in the pipeline is *JointGermlineCNVSegmentation,* which combines the gCNV segments and calls across samples to more accurately identify artifacts if found in too many control samples.

#### 2.4.2. Pharmacogenetic Haplotypes (Star Alleles)

Through targeting characterized variants in genes involved with drug metabolism, we can predict an individual’s response to a particular medication. Phosphorus maintains a database of functional haplotypes (star alleles) with clinical (disease and drug) associations, in which each star allele is defined as a unique set of mutations on a single chromosome within a designated star allele gene [14,15] (Figure 1A). The star allele pipeline examines the results of each underlying mutation within each star allele, as determined from a sample’s VCF, to assess all possible star allele diplotype results for each sample on each star allele gene. Due to the lack of phasing information available within a VCF file, there are often multiple possible valid diplotypes for a sample on a single star allele gene; in these cases, all valid diplotypes for the gene are reported for the sample (Figure 1B). All results are reviewed during QC to ensure that each underlying variant had sufficient coverage to support the call made by the VCF; Sangers are run for any variants that do not meet the minimum criteria.

#### 2.4.3. CYP2D6

Mutations in the *CYP2D6* gene are identified by the external tool Aldy [16] to accurately call complex star alleles found in the gene (Figure 2). The Aldy computational tool performs allelic decomposition and copy number detection of highly polymorphic, multi-copy genes in large sequencing sets using whole or targeted genome sequencing data. Aldy functions using a combinatorial optimization framework identifying novel and rare alleles for several important pharmacogenes, which phosphorus uses to produce the *CYP2D6* genotypes for patients. Aldy identifies all structural alteration breakpoints and reconstructs the sequence content, while considering micro structural alterations and SNVs, fusions, and hybrids among all homologous pseudogenes.

Aldy inputs high-throughput sequencing data (SAM/BAM), as well as a database of information about the genotyped gene (location of the gene, location of pseudogenes, intron/exon boundaries, and a list of all known major and minor star alleles with a unique set of mutations/structural variations). In our implementation of the Aldy pipeline on *CYP2D6*, each sample is matched against a pre-defined control sample with *2/*2 × 2 diplotype to produce a predicted *CYP2D6* diplotype for the sample. Each sample’s diplotype results are then stored in the database in the same manner as other star allele diplotypes [17].

#### 2.4.4. HLA

The human leukocyte antigen (HLA) system is a highly polymorphic system of genes. With over 7300 different HLA-I and 2200 HLA-II alleles, genotyping HLA alleles is very challenging. In addition to this vast allelic variation, HLA alleles have a high sequence similarity even across different loci, which increases the complexity of uniquely identifying a genotype using short-read sequencing techniques. We utilize two pipelines for HLA type inference: HLA I and HLA II. HLA I relies on the external software OptiType version 1.3.3 and is used to detect *HLA-A*, *HLA-B*, and *HLA-C* [18]. Reads in the FASTQ files are filtered using RazerS 3 version 3.5.3 to produce BAM files13. Next, each BAM file is converted back into a FASTQ file, where the OptiType software is run against each pair of filtered FASTQ files. HLA II runs the external software HLA*LA against each sample’s BAM file and is used to detect *HLA-DPB1*, *HLA-DQA1*, *HLA-DRB1*, and *HLA-DQB1* [19].

The HLAI OptiType program functions by selecting the combination of alleles which have the most reads with the least mismatches compared to any other allele. HLAII uses a program which uses a graph-based method with high accuracy on exome and low-coverage whole genome sequencing data [19]. To select the genotypes, the program starts with the identification of linear alignments between the input reads and the reference haplotypes from reference HLA sequences, stored in the ImGT database that a graph is constructed from; the sample’s reads are then projected onto the graph and optimized in a stepwise process specific to Population Reference Graphs [20]. The projected original linear alignment will often be a close approximation to the best graph alignment, except for reads that switch between divergent reference haplotypes.

HLA allele nomenclature is presented as Field_1*Field_2:Field_3 with the corresponding fields representing the specific HLA gene, the allele group, and the amino acid change, respectively (Figure 3).

## 3. Results

This study describes an establishment of performance characteristics for an NGS-based proactive genetics screen to detect multiple variant types, including, single nucleotide variants, indels, copy number variants, and star alleles, in 430 genes associated with adult-onset health conditions, susceptibility to various multifactorial diseases, drug response, and wellness traits. The study includes a broad range of samples representing various types of molecular results and employs in-house developed or commercially available bioinformatic tools to interrogate and account for the complexity of genetic variants. As with every laboratory-developed NGS-based test, this validation consisted of the evaluation of analytical and bioinformatics processes. Standard NGS metrics for each sample were produced and evaluated against our minimum required thresholds to have confidence in the NGS data quality (Appendix A).

### 3.1. Performance of Biological Specimen Types

The study evaluated the performance of two unpaired sample types: saliva collected in Oragene OGD-510 (DNA Genotek, Ottawa, Canada), and whole blood specimens in Vacutainer EDTA collection tubes (Becton, Dickinson and Company, Franklin Lakes, New Jersey) (Table 2). The samples were assessed based on acceptability criteria for assay quality control metrics (average depth, % of duplicate rate, % on target, alignment), as well as the accuracy of detection of previously characterized variants. Both sample types met the quality control requirements and produced expected molecular findings. The whole blood samples showed a higher average depth as compared to the saliva samples.

### 3.2. Analytical Validation

The assay was validated using three 1000 Genomes Project samples with known SNVs and indels. These samples were used to calculate replicate alignment within a single batch and across multiple batches to confirm the variant calling accuracy, as shown in Table 3a,b and Table 4a,b below. In addition, we measured the analytical sensitivity, specificity, and accuracy according to the standard formulae described in the Methods, and which are detailed in
SampleConcordantDiscordantRun 1 VariantsRun 2 VariantsRun 3 VariantsConcordance %**HG00109 ^1^**13175213481339133996.20%**HG00428 ^1^**14186014481454144295.94%**HG01350 ^1^**13577413931399139194.83%**All****4092****186****4189****4192****4172****95.65%**^1^ Each 1000 Genomes Project sample was run in triplicate.

All validation samples along with their corresponding description are listed in Appendix A. In summary, the accuracy, sensitivity, and specificity were measured at 99.92%, 97.57%, and 99.99%, respectively, for SNVs, and 94.77%, 74.57%, and 97.34%, respectively, for indels (Table 5). Lower sensitivity on indels was due to microsatellite variants in the intronic region that are difficult to detect and are not reportable. When including only reportable regions, our accuracy, sensitivity, and specificity for indels were measured at 98.98%, 94.57%, and 99.44%, respectively.

Next, we compared the frequencies across our targeted alleles to those published in the gnomAD database [12], to determine if our detection rate was in concordance with the general population. We performed a two-sample t-test to validate our concordance in each of our three targeted allele categories including susceptibility loci, wellness, and drug response. As shown in Table 6, our percent concordant was greater than 98% in each category.

### 3.3. Clinical Validation

To determine clinical sensitivity in the detection of genes with CNVs, known monogenic variants, HLA genotypes, and pharmacogenetic haplotypes (star alleles), we used 109 samples with 646 known alleles (Appendix A). The set of variants previously known for these samples is limited, so specificity could not be assessed. The analysis correctly detected 34/36 CNVs, 29/30 monogenic variants, 158/170 HLA genotypes, and 381/382 drug response genotypes (Appendix A). One known indel, c.1026_1027delAG, in the *NM_000268.3:NF2* gene was excluded from consideration as it was declared to be in a mosaic state. Another monogenic variant, in sample NA02795, was reported as pathogenic and is described as *NM_000155.3:**GALT* c.130G>A; p.Val44Met. This variant has previously been called into question as both Sanger sequencing and NGS were unable to confirm its presence [21]. Thus, this variant was removed from consideration in our validation data, resulting in a final clinical sensitivity of monogenic variants of 100% for our assay.

### 3.4. Copy Number Variants

Unlike SNPs and indels, CNVs are notoriously difficult to detect due to issues intrinsic to the technology, including short read lengths and GC-content bias. By seamlessly integrating GATK’s CNV caller into our bioinformatic pipeline, we were able to correctly identify 34 out of the 36 known genes with CNVs in our dataset (Appendix A). The CNV deletion in the *GAA* gene was excluded from the sensitivity calculation as the *GAA* gene is not covered by the assay. The two missed CNVs (*NM_004006.2:DMD*-DEL Exon 44 and *NM_004006.2*:*DMD*-DUP Exon 17) in the samples, NA04315 and NA23159, deal with single exon deletions and duplications in the *DMD* gene that are below our CNV size threshold of 200bp. Single exon CNVs present a unique challenge for tools to distinguish between true variants and coverage noise. This complication, coupled with *DMD* being a highly polymorphic gene on the X chromosome, helps to explain these two missed calls. Our final sensitivity for CNVs is 94.44% (Appendix A).

### 3.5. HLA

Test sensitivity was determined on 18 unique samples with 85 previously reported genotypes constituting 170 alleles across 7 HLA genes [22]. HLA typing results are interpreted at both a low resolution and high resolution. Low resolution results contain the first two fields, while high resolution consists of all three nomenclature fields (Figure 3). Out of 170 HLA alleles, two alleles did not match in low resolution, while 12 alleles did not match in high resolution, producing a concordance rate of 98.82% and 92.94%, respectively (Appendix A).

### 3.6. Pharmacogenetic Haplotypes (Star Alleles)

We ran our pharmacogenetic haplotype analysis on 13 well characterized genes that have been shown to affect drug metabolism: *CYP2B6*, *CYP2C19*, *CYP2C8*, *CYP2C9*, *CYP2D6*, *CYP3A4*, *CYP3A5*, *DPYD*, *NAT2*, *NUDT15*, *SLCO1B1*, *TPMT*, and *UGT1A1* [23]. In total, our assay can detect 145 unique gene haplotypes shown in Table 7. We performed both an external and internal validation process to confirm a wide range of expected results. External validation was performed on 22 Coriell samples with 334 previously characterized haplotypes [14,15,24] (Appendix A).

The concordance analysis included comparison of the expected haplotypes received based on the previously published data to the haplotypes detected in this study. However, this analysis excluded haplotypes labeled as “no consensus” and “outside haplotype detected”. No consensus indicated haplotypes previously reported as discrepant between laboratories/technologies, and thus did not produce consistent reference haplotype calls that could be used in the current analysis. Outside haplotype detected indicates a published sample haplotype consisting of star alleles that were called by other laboratories but are not covered in our assay, and thus valid comparison between the current assay and published results was not possible.

For the internal validation, a blinded study was performed with Sanger sequencing on 21 samples with 48 pharmacogenetic haplotypes (Appendix A). Only a single haplotype was missed between both studies, producing a combined sensitivity of 99.74%.

## 4. Discussion

This study describes the development and validation of a comprehensive 430-gene assay intended to screen adults for monogenic conditions and genetic susceptibilities, and to inform pharmaceutical drug selection. Although genomic-based health screening remains limited in usage, early data suggest benefits, including earlier identification in symptomatic individuals and corrected diagnoses. Further, when made aware of the opportunity for genomic screening, individuals express positive receptivity and understanding of genetic information.

The GeneCompass NGS-based assay overcomes key barriers of access, cost of goods, and utility. The GeneCompass test identifies risks and associations in multiple clinical areas influenced by monogenic diseases, complex disease susceptibility, pharmaceutical drug response, and wellness traits. This assay is custom designed to target conditions with significant medical implications, particularly those for which the risk may be mitigated through medical intervention. For example, it includes genes associated with conditions for which preventative measures have been established by, among others, the National Comprehensive Cancer Network and the American Heart Association [25,26]. It also includes the 73 genes most recently identified by the American College of Medical Genetics and Genomics (ACMG) to be reported in the case of incidental findings, due to their deleterious impact and actionability [27]. The pharmacogenetics assessment inclusion in an adult health screen is also warranted. Based on a national Dutch study, an actionable drug–gene interaction was discoverable for nearly one-fourth of new drug prescriptions [28].

Initial evidence from population-based genomic screening suggests benefits worthy of its expansion. One study demonstrated a significant finding regarding the implementation of population-based screening [29]. One laboratory reported that, among 10,478 individuals screened for up to 147 genes, 15.5% had an actionable monogenic disorder identified via the panel, including hereditary cancer and cardiovascular conditions. Only 5.2% of these were from the “ACMG 73” gene list, and only 2% were CDC Tier 1 conditions. This demonstrates the missed cases of actionable findings that occur with use of limited size panels.

Given that the limited amount of data available on population-based health genetic screens demonstrates healthy individuals receive pathogenic results, it is important to recognize that these patients were otherwise unlikely to identify their risk for disease had they not had genetic screening. Zawatsky et al. [30] reported that 76.3% of the individuals who carried pathogenic variants were unaware of their disease risk. Despite not being aware of their risk for disease, patients are still interested in identifying their risk for disease and multiple studies demonstrate that they wish to be alerted to genetic findings of medical importance.

NGS technology is applied across the GeneCompass panel, including for analysis of pharmacogenetic variants. Widespread use of pharmacogenetic analysis has historically been limited, however NGS technology has been used in research studies related to pharmacogenomics (PGx) for a decade [17]. Challenges associated with use of NGS on a broader scale for PGx testing, like identification and management of variants of uncertain significance (VUS), poor haplotype phasing, lack of appropriate software, and insufficient coverage of certain regions, are all addressed via the GeneCompass test [17]. Researchers have identified that logistics and cost-effectiveness can be optimized through a panel approach, which is best supported by clinical decision support [31]. By providing genetic counseling to patients, the option for clinical decision support is provided in real-time by a genetic counselor. While NGS presents the challenge of identifying variants of uncertain significance, this is circumvented by reporting on the PharmGKB and Clinical Pharmacogenetics Implementation Consortium (CPIC) characterized variants.

An ACMG statement describes several points to consider when implementing DNA-based screening [31]. It articulates the current knowledge limitations regarding the natural history of many genetic conditions, including their expressivity and penetrance when identified in asymptomatic individuals tested without a clinical indication. To address this, prior to receiving the GeneCompass test, patients complete a questionnaire regarding their personal and family medical history. This is reviewed by a provider network to ensure that the test is appropriate for the patient. This step, in addition to educational information provided via the company website, serves as a pre-test informational step in lieu of pre-test genetic counseling. If patients are interested in pre-test genetic counseling before moving forward with a test, sessions can be made available via a Phosphorus genetic counseling partner. When testing is completed, it is important that results be accompanied with access to genetic counseling and health care providers, to contextualize genetic findings with clinical correlates and family history.

The genetic counseling component of the process allows for patients to gain an understanding of their results, as well as the implications for their future health risks and their family members’ risk for genetic disease. Clinical action plans provided by the genetic counselor after the session further cement the understanding and direct management moving forward for both the patient and the rest of their clinical care team. Therefore, while there are bounds to the predictive value of certain genetic results, the suggestion to ensure the test content and the results are connected to expert organization and clinical follow-up is followed.

Additionally, the screening for adulthood conditions may pose a range of ethical issues, even though the test presented in this study is only offered to individuals at or older than 18 years old. The healthy screening using multi-disease panels may uncover a predisposition to an adult-onset condition in an asymptomatic individual. This may open questions on the right timing to initiate the testing to ensure the highest clinical benefits, post-test clinical management, as well as the psychological burden caused by positive test results, especially for individuals that will remain asymptomatic [32,33]. The confidentiality of the testing results, and fear of the potential impact on an individual’s insurance are also important factors to consider. Therefore, it is critical that the information about the test is explained so an individual can understand these challenges in order to make an informed decision on whether to undergo the testing [34,35].

Population-based genomic screening, still in its nascent phase, may confer benefits to broader populations than the status quo. Continued research on the clinical utility and outcomes, patient acceptability and comprehension, among other measurements are needed. Comprehensive genomic screening with lowered technical and labor costs, as achieved here, opens access to individuals outside of select health systems or research studies and increases equity in health outcomes.

## Figures and Tables

**Figure 1 jpm-12-00667-f001:**
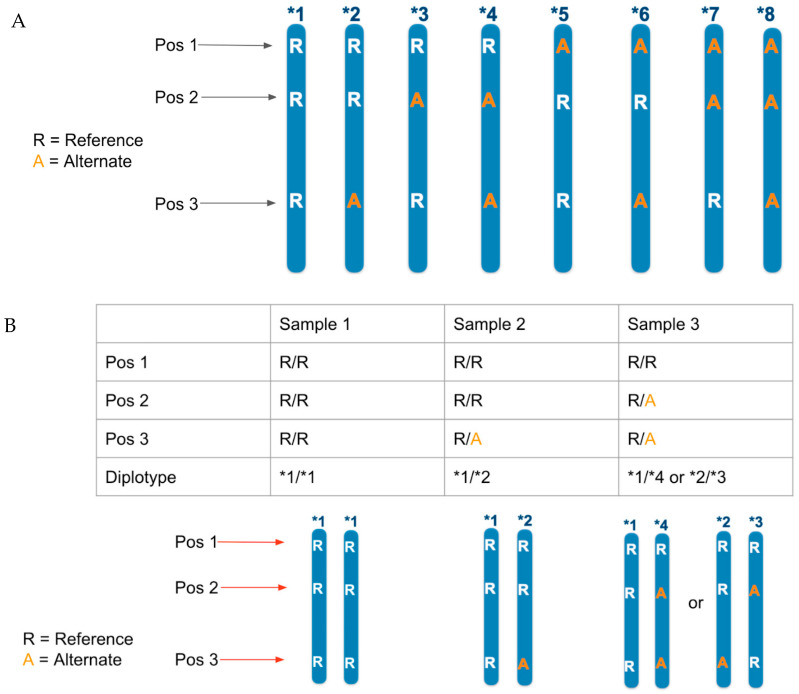
Description of multi-variant star alleles. (**A**) Possible star allele haplotypes at a genomic region. (**B**) Possible star allele diplotypes at a genomic region.

**Figure 2 jpm-12-00667-f002:**
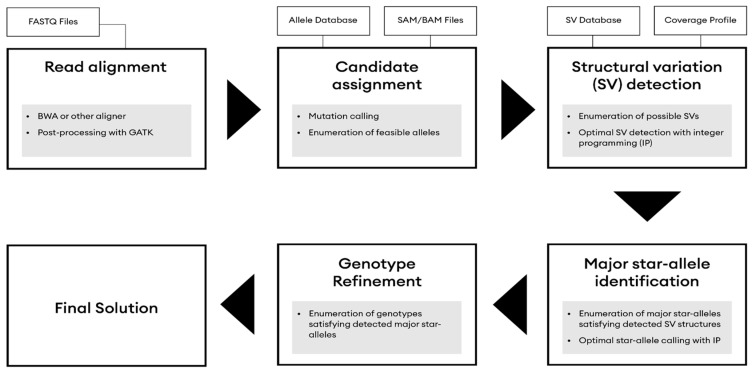
Aldy’s star allele haplotype calling workflow for variants in CYP2D6.

**Figure 3 jpm-12-00667-f003:**
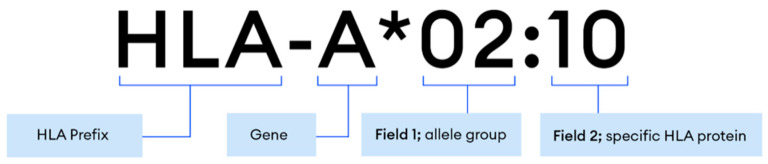
Proper nomenclature for HLA typing results.

**Table 1 jpm-12-00667-t001:** GeneCompass clinical panel content.

	Monogenic Conditions		Susceptibility Loci		Pharmacogenetic Loci
	Genes (*n*)	Conditions (*n*)	Genes (*n*)	Conditions (*n*)	Genes (*n*)
Cardiovascular	90	46	8	4	33
Endocrinology	4	2	8	4	35
Gastroenterology	2	2	9	4	5
Hearing	13	5	4	4	0
Immunology	1	1	3	3	47
Infectious Disease	0	0	8	9	18
Metabolic	14	9	0	0	0
Neurology	25	13	11	7	11
Oncology	62	51	8	6	51
Ophthalmology	10	9	4	1	24
Pain Management	3	4	2	2	17
Psychiatry	0	0	9	5	21
Pulmonology	1	1	1	1	5
Reproductive Health	39	21	9	5	40

**Table 2 jpm-12-00667-t002:** Comparison of quality metrics in blood versus saliva specimens.

Sample Type	Uniformity %	Aligned %	Duplicate %	On Target %	Total Reads	Avg. Base Depth	Base Depth ≥20×
Blood (*n* = 15) ^1^	94.33%	98.33%	1.49%	42.60%	1,338,982.00	46.96×	88.67%
Saliva (*n* = 11) ^1^	93.81%	97.29%	1.26%	43.72%	1,418,766.73	51.18×	88.91%

^1^ Averages of the two sample collection types are reported here.

**Table 3 jpm-12-00667-t003:** (**a**) Intra-precision study results; (**b**) Intra-precision variant concordance.

(a)						
Sample	Total Reads	Average Depth	% Aligned		% Duplicate Reads	% On-Target
HG00109A	2,433,884	86.15	98.64%		3.82%	43.57%
HG00109B	2,119,180	106.02	99.13%		3.05%	61.59%
HG00109C	2,174,032	108.26	99.21%		2.57%	61.30%
Average	2,242,365	100.14	98.99%		3.15%	55.49%
HG00428A	2,675,926	95.56	98.36%		4.05%	43.96%
HG00428B	2,152,570	107.84	98.67%		3.59%	61.67%
HG00428C	2,204,744	108.02	99.17%		2.50%	60.31%
Average	2,344,413	104.00	98.73%		3.38%	55.31%
HG01350A	2,694,592	93.64	98.61%		4.11%	42.78%
HG01350B	2,418,592	121.57	99.08%		3.72%	61.88%
HG01350C	2,739,276	130.64	98.99%		2.55%	58.71%
Average	2,617,487	115.00	98.89%		3.46%	54.46%
**(b)**						
**Sample**	**Concordant**	**Discordant**	**Replicate A Variants**	**Replicate B Variants**	**Replicate C Variants**	**Concordance %**
**HG00109A/B/C**	1329	65	1375	1360	1361	95.33%
**HG00428A/B/C**	1426	85	1476	1464	1463	94.37%
**HG01340A/B/C**	1368	82	1412	1404	1422	94.34%
**All**	**4123**	**232**	**4263**	**4228**	**4246**	**94.67%**

Each 1000 Genomes Project sample was run in triplicate.

**Table 4 jpm-12-00667-t004:** (**a**) Inter-precision study results; (**b**) Inter-precision variant concordance.

(a)						
Batch	Sample	Total Reads	Average Depth	% Aligned	% Duplicate	% On-Target
Run 1 Operator 1	HG00109	2,240,126	116.88	99.20%	3.14%	64.23%
	HG00428	2,198,956	114.36	99.22%	3.08%	64.02%
	HG01350	1,939,004	100.91	99.24%	2.47%	64.07%
	Average	2,126,029	111.00	99.22%	2.90%	64.11%
Run 2 Operator 1	HG00109	2,433,884	86.15	98.64%	3.82%	43.57%
	HG00428	2,675,926	95.56	98.36%	4.05%	43.96%
	HG01350	2,694,592	93.64	98.61%	4.11%	42.78%
	Run 2 Average	2,601,467	91.78	98.50%	4.00%	43.40%
Run 3 Operator 2	HG00109	2,059,476	105.68	99.22%	3.06%	63.17%
	HG00428	2,059,476	105.68	99.22%	3.06%	63.17%
	HG01350	2,189,582	113.96	99.03%	2.82%	64.07%
	Run 3 Average	2,102,845	108.00	99.00%	3.00%	63.00%
**(b)**						
**Sample**	**Concordant**	**Discordant**	**Run 1 Variants**	**Run 2 Variants**	**Run 3 Variants**	**Concordance %**
**HG00109**	1317	52	1348	1339	1339	96.20%
**HG00428**	1418	60	1448	1454	1442	95.94%
**HG01350**	1357	74	1393	1399	1391	94.83%
**All**	**4092**	**186**	**4189**	**4192**	**4172**	**95.65%**

Each 1000 Genomes Project sample was run in triplicate.

**Table 5 jpm-12-00667-t005:** Accuracy, sensitivity, and specificity.

	Reportable Regions		All Regions	
	SNV	Indel	SNV	Indel
True positives	10,164	689	10,164	689
True negatives	366,657	7073	366,657	7073
False positives	10	53	53	193
False negatives	54	43	253	235
Accuracy	99.98%	98.78%	99.92%	94.77%
Sensitivity	99.47%	94.13%	97.57%	74.57%
Specificity	99.99%	99.26%	99.99%	97.34%
Positive predictive value	99.90%	92.86%	99.48%	78.12%
Negative predictive value	99.99%	99.40%	99.93%	96.78%

**Table 6 jpm-12-00667-t006:** Sample Allele Frequency vs. Population Allele Frequency.

Targeted Category	Gene Count	T-Test Concordance %
Disease Susceptibility	62	98.80%
Wellness	91	99.48%
Drug Response	81	99.81%

**Table 7 jpm-12-00667-t007:** Pharmacogenetic haplotypes detected by assay.

Gene	Haplotypes Detected
CYP2B6	*4, *6, *9, *18, *22, *34, *36
CYP2C19	*1, *2, *3, *4, *5, *6, *7, *8, *9, *10, *13, *17
CYP2C8	*1A, *2, *3
CYP2C9	*1, *2, *3, *4, *5, *6, *7, *8, *9, *10, *11, *12, *13, *15, *16
CYP2D6 ^1^	*1, *1 × 2, *1 × 3, *1 × 4, *2, *2 × 2, *2 × 3, *2 × 4, *3, *4, *4 × 2, *5, *6, *7, *8, *8 × 2, *9, *9 × 2, *10, *10 × 2, *11, *12, *13, *14, *15, *17, *17 × 2, *21, *29, *29 × 2, *35, *35 × 2, *36, *40, *41, *41 × 2, *45, *49, *50, *54, *55, *56, *59, *61, *61 × 2, *62, *63, *65, *68, *68 × 2, *69, *72, *74, *82, *83, *83 × 2, *84, *96, *109, *114
CYP3A4	*1, *2, *13, *15, *22
CYP3A5	*1, *2, *3, *3B, *6, *7
DPYD	*1, *2A, *6, *9A, *13, rs67376798
NAT2	*4, *5, *6, *7, *14
NUDT15	*1, *2, *3, *6, *9
SLCO1B1	*1A, *1B, *5, *15, *17, *21
TPMT	*1, *2, *3A, *3B, *3C, *4, *8, *24
UGT1A1	*1, *6, *28, *36, *37, *80

^1^ Additional structural variants may be detected, including hybrid and tandem alleles.

## Data Availability

Not applicable.

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
