# Peer review of "Technical Performance of a 430-Gene Preventative Genomics Assay to Identify Multiple Variant Types Associated with Adult-Onset Monogenic Conditions, Susceptibility Loci, and Pharmacogenetic Insights"

_jpm, 2022, doi:10.3390/jpm12050667_

Round 1

Reviewer 1 Report

The present article describes an NGS commercial panel for the screening of pathogenic monogenic disease variants, genetic risk factors, pharmacogenetics and HLA haplotyping, and wellness traits. It presents a technical and clinical validation, based on the analysis of several commercial and internal controls for each of the variant types. One or some thoroughly sequenced controls like Genome in a bottle NA12878 would add much more information with a low sequencing cost, and would guarantee an unbiased selection of test variants. Some metrics common to NGS validation studies are absent.

The actual target for each gene is not completely explained and the read depth obtained is rather low if we compare it with diagnostics gene panels. More than 11% of the target region is undercovered (read depth below 20x), but maybe for a general screening panel it can be accepted, provided that the sensitivity limitations are well explained in the report. I have some doubts about the sensitivity/specificity calculations; therefore some specific questions are made below to ensure proper results.

I’m not an expert in pharmacogenomics or HLA haplotyping, so my ability to judge if this panel can reliably genotype this type of variation is limited, especially to know if the targeted alleles cover all the clinically relevant ones, or if the uncertainty for some diplotypes due to short red technology severely limits its utility.

Specific comments:

- Post-test genetic counselling is well discussed. Some words about the benefits of pretest counselling would be advisable too.

- Variant nomenclature: All variants, or at least point variants, should be described in an additional column with the HGVS nomenclature, including the reference sequence used. Exon deletions/duplications should be described accompanied of the reference transcript for  exon numbering. This includes Tables S1, S3 and S6.

- Uniformity of coverage: this key value is not described in the article. It could be defined as the percentage of targeted base positions in which the read depth is greater than 0.2 times the mean region target coverage depth (Pct > 0.2*mean). It would be advisable to include it.

- Intra- and inter-precision measurements are presented as a table of some sequencing metrics for different samples or runs. I think that presenting the average of three runs in each row of the inter-precision table masks the inter-run differences and each sample-run pair shoud be shown separately, or a measure of the variance should be presented. Furthermore, no measure on the actual variant calling differences is presented. Repeatability (the percent agreement in variant calling between successive tests carried out under the same conditions of measurement) and reproducibility (the percent agreement between the variant calling of tests carried out under a variety of conditions (e.g., different operators, machines, and time frames)) calculations should be also too.

- Sensitivity/specificity calculations: In Table S3, variant NF2 c.1026_1027delAG is dismissed with the comment “Outside Interval Range”, allowing a 100% sensitivity value for point variants. However, being NF2 a tumor suppressor gene, from the monogenic type list and according to the M&M section, coding+-12bp regions should be included in the target region. This variant is in the coding region, specifically in an exon where ClinVar describes frameshift and nonsense variants as pathogenic. This region being considered outside interval range should be discussed in the article. Also the CNV consisting in the deletion of exons 7-15 of GAA is not counted for the “concordance” numbers for being “Outside Interval Range” and no mention is made in the article. Again is a region filled with clinical relevance. The proportion of coding+-12bp regions in “monogenic type” genes considered “outside interval range” should be quantified. This should also be mentioned in the limitations section of the genetic report.

- There is no clear technical validation of CNV calling. Sensitivity of CNVs is described in the text as the result of finding 33 out of 35 variants, and Table S6 is cited. However, according to this table 34 out of 36 variants are detected. Please correct the discordance. Furthermore, the article claims that the test was able to correctly identify the CNVs, but it should state that the test detected variants in genes containing CNVs, clarifying that the variants were detected at the gene level, but in many cases the specific exons altered were not correctly determined. Some CNVs produce in-frame non-pathogenic deletions or insertions, so the specific altered exons can be decisive. Hopefully in the report there will be an explanation of the low accuracy of the actual altered exons and the need of further assays to determine the pathogenicity of the variant.

- Star allele diplotyping: What does “no consensus” and “outside haplotypes detected” exactly mean in Table S4a? Some explanation of why those cases are dismissed would be welcome, and would help understand how the totals are counted.

- Explain better “Pathogenicity assessment for variants identified in genes tested for monogenic disorders was assessed based on the American College of Medical Genetics (ACMG) guidelines and internal variant curation laboratory protocol.”  Which do you use in what cases or how do you combine them? Do you use only the Richards et al (PMID 25741868) primary guidelines or do you incorporate the ClinGen updates and gene-specific guidelines?

- I do not see the importance of figure 3, to be included as a main figure, but .

- The article reads “It also includes the 73 genes most recently identified by the American College of Medical Genetics and Genomics (ACMG) to be reported in the case of incidental findings, due to their deleterious impact and actionability [27]” I find this sentence misleading, since the cited list (most recent version) includes 82 genes, so 9 are missing. Please change it to “It also includes the 73 of the 82 genes…”, “most of the 82 genes…” or something similar.

- Line 403 cites tables S3-S7, but table S7 is not provided. Please correct it.

Reviewer 2 Report

The manuscript entitled "Technical Performance of a 430-gene Preventative Genomics 2 Assay to Identify Multiple Variant Types Associated with 3 Adult-Onset Monogenic Conditions, Susceptibility Loci, and 4 Pharmacogenetic Insights" presents a gene panel of genetic screening in healthy individuals. The manuscript is well-written and the methodology applied to the NGS assay was finely constructed. My main concern is the lack of a strategy to explain how all these variants will be interpreted in the clinical scenario. In addition, some comments I would like to address to the authors.

  1. I did not encounter the Supplemental Material; hence, I would like to evaluate the genes selected for further review.
  2. The order of the Tables is incorrect. Table 6 appears in the text right after Table 1. The same problem happens to the Supplemental Material. Some genes are not italicized as well.
  3. In lines 54-55, the authors mention that ACMG lacks endorsement regarding genetic screening in healthy people. Although the ACMG guideline is cited again in the Discussion section, I believe this affirmative sounds accusatory, even if non-intentionally.
  4. The authors justify their strategy because of the individual's wish to know more about their genome. I believe the bioethics concerns regarding genetic screening in healthy individuals should be better addressed. Will be this genetic screening available to any individual? Is there genetic counseling available for everyone? In addition, the strategies for VUS interpretation, phenotype modifiers, de novo variants, functional validation, etc., should be mentioned and properly discussed.

Round 2

Reviewer 1 Report

First I apologyze for a short delay in the review; for the second time supplementary tables were not available and I had to ask for them to the editors.

Dear authors, thank you for the corrections and additions made.

However, there are still some comments that have not been completely addressed:

-Variants described in supplementary tables still lack the associated reference transcript. It is true that the Coriell repository do not provide it always, but since most of the variants have been found, the reference sequence should be known, at least in them. Transcripts are only present in the panel gene description, but probably do not match always with that used for the Coriell variants. I insist that technical articles should always describe variants properly.

-A Uniformity column has been added to Table 2, but I cannot understand how has it been calculated: If for blood the average depth is  98x and about 89% of regions have coverage >=20x, the uniformity, being the percentatge of bases with coverage >= 19.6x (98*0.2), cannot about 98.9%. It also appears in the new table S7, where strikingly the asked parameter seems to be the last column with another name. What is then Uniformity Pct? Since uniformity is now defined in the article, the number should match the definition.

-Now the same table appears in lines 392 and 407.

-Regarding the ACMG list of “must report” incidental findings, I am sorry for my mistake: the list has more  lines than genes, since some genes appear repeated. The autors were right about it including 73 genes.

Reviewer 2 Report

The authors have performed the alterations I have requested and I believe the manuscript is now suitable for publication in the Journal of Personalized Medicine.

Author Response

We appreciate the reviewer's time and feedback on this manuscript.